Geology correlates with gut microbial community composition in the Mountainsnails (Oreohelicidae: Oreohelix)

Oiler Ian M. oile6353@vandals.uidaho.edu ian.oiler@rutgers.edu 1 2
Linscott T. Mason 3
Parent Christine E. 1 2
1 Department of Biological Sciences, University of Idaho , Moscow , ID , United States of America
2 Institute for Interdisciplinary Data Sciences (IIDS), University of Idaho , Moscow , ID , United States of America
3 Department of Biological Sciences, Virginia Polytechnic Institute and State University (Virginia Tech) , Blacksburg , VA , United States of America
Sunny Armando
Electronic publication date: 2025 Oct 7
Publication date: 2025
Volume: 13
Electronic Location ID: e20080
Received 2024 Sep 12; Accepted 2025 Aug 24
Copyright: ©2025 Oiler et al.
Copyright year: 2025
Copyright holder: Oiler et al.
License: This is an open access article distributed under the terms of the Creative Commons Attribution License, which permits unrestricted use, distribution, reproduction and adaptation in any medium and for any purpose provided that it is properly attributed. For attribution, the original author(s), title, publication source (PeerJ) and either DOI or URL of the article must be cited.
License URL: https://creativecommons.org/licenses/by/4.0/

Keywords: Microbiome, Mollusk, Gastropod, Gut, 16S rRNA, Next-generation sequencing, Geology, Calcium carbonate

Funding: The National Science Foundation NSF 1751157 NIH COBRE P30GM103324 This work was supported by a CAREER grant from the National Science Foundation (NSF 1751157) to Christine E. Parent. Data collection and analyses performed by the IIDS Genomics and Bioinformatics Resources Core at the University of Idaho were supported in part by NIH COBRE grant P30GM103324. The funders had no role in study design, data collection and analysis, decision to publish, or preparation of the manuscript.

==============================
Background

Species that require soil mineral macronutrients for survival may depend on specific microbiome communities to aid in nutrient processing. Land snails, which utilize environmental minerals to synthesize a shell of calcium carbonate (CaCO3), may rely on or possess distinct gut microbiome communities depending on soil mineral characteristics. Here, we investigate whether the occurrence of calcareous vs. non-calcareous soils is associated with shifts the composition of the gut microbiome of the calciphilous and highly diverse land snail genus Oreohelix from the Western United States.

Methods

We collected snail and soil samples from nine sites in central Idaho: five near, and four away from calcium-rich geology. We sequenced the V4 region of the 16S rRNA gene of these samples to assess the gut microbiome compositions of Oreohelix land snails on and off calcium-rich substrates. After data clean-up and filtering we had 68 snail and 25 soil microbiome samples.

Results

We found that snail gut microbiomes differed significantly from the surface soil microbiome, with many amplicon sequence variants being unique and ubiquitous in the snails. We also found small, but significant, differences between snails on and off calcium-rich rocks. Our findings indicate that the gut microbial community assembly process of land snails is complex and does not reflect a simple relationship with the underlying soil microbiome. While we find a pattern of differences associated with the proximity of calcium-rich geology, the snail microbiome communities are likely forming based on a variety of other factors, including diet and host filtering. Furthermore, we found multiple microbial taxa that were ubiquitous in the snails and rare in the nearby substrate microbiomes. Future work should focus on disentangling the role of habitat and the functional importance (or lack thereof) of the microbial taxa that are common to almost every sampled snail.

Introduction

Gut microbes can affect their hosts in diverse ways, ranging from benefits such as increased fitness, to neutral effects with no apparent impact, to detrimental outcomes that reduce fitness. Similarly, host species can vary in their dependence on gut microbes, with some species completely depending on certain microbes for survival (Ley et al., 2008; Bourguignon et al., 2018) and others able to survive without any gut microbiome at all (Hammer, Sanders & Fierer, 2019). Each system of microbial ecology falls somewhere along this spectrum from no effect to strong commensal or mutualistic effects—though we still do not know the distribution nor the environmental or host factors that contribute to where microbiomes fall on that spectrum. While much focus has been devoted to human and other mammal microbiomes, recent efforts have been made to increase the taxonomic breadth of hosts considered in microbiome studies (Pascoe et al., 2017). In addition, expanding the taxonomic scope of hosts in the study of gut microbiomes may help accomplish a major goal of microbial ecology: to predict the distribution and types of microbes that occur in different environments. Central to this aim is to increase the breadth of studied host species to understand how differences in host filtering (whereby factors such as gut pH or the immune system constrain microbial colonization or growth) and host environment interact and shape the composition of microbial communities.

Many factors can affect the composition of microbial communities in a given environment. For gut microbiomes, factors like diet, gut pH, and host evolutionary divergence, measured by phylogenetic relatedness, have been found to be important drivers of gut microbiome differences (Ley et al., 2008; Beasley et al., 2015; Moeller et al., 2016). Thus far, much of this research has been done in humans and non-human mammals. However, steady progress is being made to narrow the gap between our understanding of mammalian and non-mammalian gut microbiome dynamics. For example, some studies have shown that factors such as host phylogenetic relatedness are stronger predictors of vertebrate (particularly mammalian) microbiome similarity than for invertebrate hosts (Song et al., 2020; Hammer, Sanders & Fierer, 2019), showing that some trends might be specific to certain clades. Thus, broadening the diversity of taxa and considering a wider range of important factors that shape microbiomes is critical to understanding how diverse microbial communities assemble. This will eventually allow for the prediction of microbial community patterns and responses across the tree of life.

Despite their ecological and taxonomic diversity, terrestrial molluscs are poorly represented in microbial literature. Land snails are adapted to a variety of habitats and food sources and include species that are herbivorous (Lissachatina fulica; Cardoso et al., 2012), predatory (Oxychilus alliarrus; Curry & Yeung, 2013), and detritivorous (Oreohelix; Bernard & Wilson, 2016). Each of these diets may come with a different level of reliance on microbes in their gut, and therefore, different microbial community compositions and structures. More generally, the resources that animals need access to or must metabolize can have large effects on whether their microbiome is composed of obligatory symbionts or mostly transient microbes (Hammer, Sanders & Fierer, 2019). Common examples include animals that consume fresh vegetation (including land snails; Pinheiro et al., 2015), as they often rely on gut symbionts to digest complex macromolecules like cellulose and lignin. In land snails we can contrast this dietary reliance on microbes with taxa that are microbial grazers and share much of their microbiome with their immediate environment (O’Rorke et al., 2015). Recent work in microbial ecology has explored how host traits shape microbial community assembly through processes such as host filtering (e.g., Mazel et al., 2018). In the present study, we utilize a known association between Oreohelix land snails and the presence of calcium-rich rocks in their environment to identify these types of community assembly patterns.

The Mountainsnails, genus Oreohelix, are a highly threatened group of land snails that occupy diverse geologic substrates and have only recently begun to be studied for their microbial associations (Chalifour & Li, 2021; Chalifour, Elder & Li, 2022; Chalifour, Elder & Li, 2023). Research has thus far shown three major factors shape microbial composition in some species of Oreohelix: (1) vertical transmission of microbial components between parents and offspring (Chalifour & Li, 2021), (2) gut microbiomes of Oreohelix strigosa are not significantly different between recent and decades old museum samples (Chalifour, Elder & Li, 2022), and (3) geography (defined in the present study as spatial distances between samples) is a major driver of gut microbiome variation across the distribution of O. strigosa (Chalifour, Elder & Li, 2023). It has also been shown that some select microbial families often dominate these snails’ gut microbial communities (Chalifour, Elder & Li, 2023). Together, these results suggest host filtering and/or a symbiotic relationship between Oreohelix land snails and their gut microbiomes, which highlight potential effects from local ecological and environmental conditions that influence microbiome composition.

Land snails exist in a diverse set of environments and tap into many different resources to acquire calcium carbonate (CaCO3), essential to build their shells. Specialized microbiome communities may enable land snails to access these resources. Additionally, in environments rich in CaCO3, bacteria in the Oreohelix gut might be exposed to a higher pH due to the liming properties of CaCO3. Previous research found that Oreohelix populations that live on CaCO3–rich rock outcrops (like limestone and marble) have increased biomineralization expression relative to non-CaCO3 rock resident populations (Linscott, Recla & Parent, 2023). Notably, CaCO3 is also known to be a common agricultural soil amendment to manipulate pH levels leading to the increase of microbial abundance and/or diversity (Fierer & Jackson, 2006). Our study addresses how the association between snail geographical distribution and the presence of CaCO3-rich rocks might be associated with the gut microbiome composition of Oreohelix land snails. For example, this rock-microbiome association could be the result of rock-specific soil microbial communities associated with the CaCO3-rich rock substrate or of specific snail gut microbes associated with the ingestion and processing of CaCO3-rich substrate by the snails. Since geology (in the current study defined as presence of CaCO3-rich rocks) and its potential effect on microbial soil composition can be a major factor driving Oreohelix distributions and shell biomineralization (Crowther et al., 2019), gut microbiome communities of Oreohelix host snails on CaCO3 rock and non-CaCO3 rock may be distinct. Thus, understanding the effect of geology on the variation in gut microbiomes of Oreohelix land snail hosts has the potential to shed light on the association between a given host, the community composition of its microbiome, and the environment where it lives. Previously untested, such geologic driven differences in host microbiota may have implications for microbial assembly patterns in other host taxa that depend on specific geological properties.

In this study, we test for the effects that host filtering and geology have on Oreohelix land snail microbiome composition by comparing soil and snail bacterial communities associated with CaCO3 rock and non-CaCO3 rock habitats. To do so, we amplified and sequenced the V4 region of the 16S rRNA gene from snail and soil microbiome samples to test the effect of multiple environmental variables including geology, host species, and geography, on host and soil microbiome composition to better understand how CaCO3 availability may be related to land snail gut microbiome composition.

Materials and Methods

Sampling

We collected a total of 69 snail samples and 27 soil samples in July 2020, when snails were known to be active at the selected sampling locations. We selected sites roughly between Lewiston and Riggins, ID based on geological characteristics, using geologic maps and previous studies (Kauffman et al., 2014; Linscott, Recla & Parent, 2023) to characterize sites as either being on or off calcareous rock (i.e., CaCO3 rock). Of the sites we chose, five were on and four were off CaCO3 rock (Fig. 1).

At each site, we collected three surface soil samples and a minimum of three and up to ten live snails (mean number of samples per site = 7.66) opportunistically and placed each individual into sterile vials. We transported the samples to the University of Idaho where soil samples were stored at −80 °C and snails at room temperature. The snail species and subspecies sampled are morphologically and geographically distinct, and their identities were confirmed in prior work using the same populations (Linscott, Recla & Parent, 2023). Snails in our study, except for O. jugalis (site LUC01), include those in the recently diverged O. strigosa species complex (Linscott et al., 2020). A pruned tree based on Linscott et al. (2020) is presented in Fig. 1 to show the phylogenetic relationships among species/subspecies in the present study. None of the sampled populations occur in sympatry save O. jugalis and O. idahoensis which can sometimes be found in sympatry at the geologic boundary between volcanic and limestone populations.

Sample processing

We sacrificed live snails within two days of collection, removed them from their shells, and gut tissue (intestine from digestive gland to anal pore) was dissected, rinsed with sterile water, and stored in 95% ethanol at −20 °C until extraction. Our extractions of snail (whole gut tissue) and soil (approximately 100 mg of bulk soil per sample) DNA were conducted using the DNeasy Powersoil Pro Kit (Qiagen) using the manufacturer’s instructions. We eluted extractions into 50 µl of C6 solution and stored them at −20 °C until library preparation.

Figure 1 Sampling locations with site names and matching phylogeny.

Sites in light blue represent those on calcium carbonate-rich rocks (typically limestone), while darker blue represents those without calcium carbonate-rich rocks nearby. All taxa sampled, except O. jugalis, are part of the O. strigosa species complex.

Library preparation and sequencing

We amplified the V4 hypervariable region of the 16S rRNA gene and barcoded it using the 515F/806R primer pair and protocol from the Earth Microbiome Protocol for 2-step PCR (Thompson et al., 2017). Positive and negative controls were used at each step and progress was checked using a Qubit Fluorometer (ThermoFisher Scientific) and gel electrophoresis. We utilized qubit scores and gel band intensity to ensure equal amounts of DNA were included in the libraries. All the libraries were sequenced on a 300 bp paired-end Illumina Miseq platform PE300 (Illumina Corporation, San Diego, CA, USA) lane by the Genomics and Bioinformatics Resources Core at University of Idaho.

Data analyses

We processed reads for further analyses using QIIME 2 2024.10 (Bolyen et al., 2019). We used the q2-demux plugin to quality filter the raw sequences, followed by denoising with DADA2 (via q2-dada2; Callahan et al., 2016). We trimmed the sequence length during DADA2 based on visual inspection of sequence quality using Phred scores while maintaining overlap between forward and reverse sequences. These thresholds were chosen as 230 bp for forward reads and 130 bp for reverse reads. Using extraction and PCR blanks, contaminants were then identified by prevalence using the Qiime2 plugins for DECONTAM (decontam-identify and decontam-remove) (Davis et al., 2018). We aligned the amplicon sequence variants (ASVs) using MAFFT (via q2-alignment; Katoh et al., 2002) and used the aligned sequences to construct a phylogeny with fasttree2 (via q2-phylogeny; Price, Dehal & Arkin, 2010). Samples below 5,000 reads were dropped (based on rarefaction curves; see Fig. S1) as well as ASVs with less than 10 reads in any sample to remove possible contaminants. This left 93 total samples (68 snails, 25 soils) with 14,184 ASVs and 3,075,768 total reads.

For alpha-diversity metrics (observed ASVs, Faith’s Phylogenetic Diversity (Faith, 1992), and Shannon’s diversity index (Shannon, 1948)), beta diversity metrics, Robust Aitchison distance (Martino et al., 2019) (using QIIME2 version 2023.5), and Principal Coordinate Analysis (PCoA) we used the plugins deicode and q2-diversity. To assign taxonomy to ASVs, we used the q2-feature-classifier (Bokulich et al., 2018) classify-sklearn using the pre-trained classifier Silva 138 99% OTU database (Robeson et al., 2021; Quast et al., 2013) trained on the 515F/806R region of the 16S gene. We used PERMDISP and Adonis (PERMANOVA) (Anderson, 2001; Oksanen et al., 2023) to compute beta-diversity differences between samples using q2-diversity beta-group-significance and q2-diversity adonis. We included presence of CaCO3-rich substrates at the sampling location (Geology) and whether samples were soil or snail derived (Sample Type) in our assessment of overall variation in beta diversity. Furthermore, within soil samples we again assessed the variation in beta diversity for geology. For snail samples, we assessed beta diversity variation for geology and snail taxa. We also assessed the beta diversity variation explained by geology within two localities (PITS and RR) that had a single species and sites on and off limestone to mitigate the effect of host taxonomy.

We migrated all output files to R using the package qiime2R (Bisanz, 2018). For soil and snail sample types, we used Mantel and partial Mantel tests with the vegan package in R (Oksanen et al., 2023) to test whether there was a correlation between microbiome composition using Robust Aitchison distance, geographic distance between samples, and/or presence of calcium-rich rocks (see Table 1 for details). Differential abundance analyses were done between sample types (snail vs soil) and between samples on/off CaCO3 within sample types, using the full dataset as well as the subset of only RR and PITS sites. We performed analysis of compositions of microbiomes with bias correction (ANCOM-BC; Lin & Peddada, 2020) at various taxonomic levels (family level reported).

Table 1 Summary of beta diversity results.

Each test shows the effects of the given variable on microbial distances, using Robust Aitchison distance. Subspecies distinctions within the O. strigosa species complex are used to for the ‘Species’ comparisons. P values are denoted by number of asterisks (*, **, or ***), depending on the level of significance.

Samples tested	Test used	Variable	Test statistics	P value	
Soil only	PERMANOVA	Geology	df = 1, F = 0.95,
R2 = 0.04	0.40	
	PERMDISP	Geology	df = 1, F = 1.04	0.26	
	Mantel	Geography	ρ = 0.35	0.001***	
	Mantel	Geology-Geography	ρ = 0.001	0.415	
Snail only	PERMANOVA	Geology	df = 1, F = 6.38,
R2 = 0.09	0.002**	
	PERMDISP	Geology	df = 1, F = 1.05	0.31	
	PERMANOVA	Species	df = 5, F = 5.27,
R2 = 0.3	0.001***	
	PERMDISP	Species	df = 5, F = 2.00	0.062	
	PERMANOVA	Species + Geology	df = 5, F = 5.46,
R2 = 0.3	0.001 (Species)***	
			df = 1, F = 3.32,
R2 = 0.04	0.029 (Geology)*	
	Mantel	Geography	ρ =  − 0.09	0.98	
	Mantel	Geology - Geography	ρ = 0.05	0.008**	
RR/PITS snails	PERMANOVA	Geology	df = 1, F = 4.73,
R2 = 0.17	0.008**	
	PERMDISP	Geology	df = 2, F = 3.03	0.077	
Soil and snail	PERMANOVA	Sample type	df = 1, F = 45.23
R2 = 0.33	0.001***	
	PERMDISP	Sample type	df = 1, F = 4.91	0.028*	

Effect size and power

We calculated effect size and power for the alpha diversity comparison between soil and snail samples and between snails on and off calcium-rich substrates. We calculated Cohen’s d for effect size using the command cohen.d in the effsize package (Torchiano, 2020) in R. Cohen’s d was used to calculate power using the pwr.t2n.test command in the pwr package (Champely, 2020).

Results

Our study looked at the effects of CaCO3 rock on Oreohelix land snail microbiome composition. We found that the snail gut microbiomes were affected by different factors than soil microbial communities and were highly differentiated from environmental soil microbiomes. While the effect size is small, we found significant differences between snail samples on and off CaCO3 rock. In addition, we found several microbial taxa that are highly abundant and unique to the snail samples.

Compositional differences between sample types

Our results showed that snail gut microbial compositions were significantly different from soil samples using Adonis (Oksanen et al., 2023) on Robust Aitchison distances (see Table 1). Using PERMdisp, we showed that there is also a difference in dispersion between sample types (F = 4.91, p = 0.028, permutations = 999). Figure 2 shows the separation of the two sample types, and the increased dispersion of snail samples compared to soil samples.

Figure 2 PCoA of Robust Aitchison distances colored by treatment group.

PERMANOVA results find significance between sample types as well as snails on and off calcium carbonate-rich rock. However, calcium carbonate-rich rock had no effect on soil sample differences. Snail samples were also found to have greater dispersion than soil samples.

Compositional differences within sample types

Within sample types, we found that soil sample differences were significantly correlated with geographic distance but not significantly affected by CaCO3 rock presence (Table 1). In snails, the reverse was true. There was no effect of geography (defined as spatial distance between sites), but CaCO3 rock presence was found to be a significant factor for distinguishing microbiomes (Table 1). We also found that, when taking geographic distance into account using partial Mantel tests, the effect of CaCO3 rock on microbiome dissimilarity remained significant (Table 1). Additionally, species identity was found to be a significant factor for snail microbiome differentiation, and geology (based on presence of CaCO3-rich rock) remained a significant effect when accounting for species effect (Table 1). To further account for the effect of taxonomy, we assessed the compositional differences within a subset of microbiome samples from the same snail species. Namely, two locations, PITS containing O. strigosa and RR containing O. intersum, have the same taxa at sites on and off limestone. Using only these, we found that the effect of being on/off limestone is still significantly correlated with difference in microbiome composition (see Table 1).

Alpha diversity

Effect size and the power to detect difference was high for Shannon’s diversity between sample types but relatively low for all other alpha diversity measures (see all values in Table S1). With this in mind, we found no significant differences in observed richness (Kruskal–Wallace, H = 2.47, p = 0.12) and Faith’s phylogenetic diversity (Kruskal–Wallace, H = 0.54, p = 0.46) between soil and snail sample types. However, Shannon’s index showed significantly higher diversity in soil compared to the snail samples (Kruskal–Wallace, H = 12.87, p < 0.001—boxplots in Fig. S2). Within sample types, we found that alpha diversity was not significantly affected by CaCO3 presence in either soil or snail samples.

Taxonomic results

We used taxonomic analyses to assess differences between samples based on microbe identity. Large differences were clear between snail and soil samples based on taxonomy (Fig. 3). For example, soil samples had significantly less of the phylum Bacteroidota and significantly more Crenarchaeota, an Archaeal phylum. In addition to having more Bacteroidota in the snail gut samples, there was also much more Deinococcota. At the family level, there was a lot of diversity, with only a few taxa being over 5% of any group of samples. The most abundant of these were Nitrososphaeraceae in soils and Spirosomaceae, Sphingomonadaceae, and Trueperaceae in snails.

Figure 3 Jitterplot depicts mean relative abundance for the five most abundant phyla.

Points indicate individual sample values and crossbars indicate median values for each sample category. Points are jittered in the horizontal direction for visibility.

To find core microbes, we looked for microbes that appeared in most snail samples. Only one ASV, family Trueperaceae (phylum Deinococcota), was found to be present in every snail sample. This ASV was also found in many of the soil samples at very low (<1%) relative abundance. In fact, nearly 85% of all samples contained this ASV. There were four other ASVs that were found in around 85% of Oreohelix snail samples. These were from the following families: Spirosomaceae, Sphingobacteriaceae, Sphingomonadaceae, and Nocardioidaceae. Spirosomaceae was the most abundant family found in the snail samples and was hardly found at all in soil samples. Sphingobacteriaceae was found at low relative abundance in snails, but even lower relative abundance in soils. Sphingomonadaceae was found in higher relative abundance in snails but was also present in most soil samples. Nocardioidaceae was found at similar relative abundances in snail and soil samples.

There were some snail samples that were dominated by outlier taxa. The main instance of this was bacteria in the family Francisellaceae, which dominated (over 20% of sample total—sometimes over 50%) the communities of samples from three disparate sites and was either absent or at much lower relative abundance in the rest of the samples.

Differential abundance

Differential abundance testing, using ANCOM-BC, confirmed the differences between sample types with many differentially abundant taxa. There were 93 microbial families that were differentially expressed between soil and snail samples (Table S2). Within snail samples, we found that there were two families, Sandaracinaceae and Hyphomonadaceae, that were differently abundant between samples on CaCO3 rock and off. Both were found to be more abundant in the off-limestone samples. For the subset dataset using only RR and PITS sites, we also looked at taxa significantly different based on CaCO3 proximity and found four significant families including Sandaracinaceae again as well as Chroococcidiopsaceae, Rubrobacteriaceae, and Pyrinomonadaceae. In soil samples, the only differently abundant families were unclassified bacteria in the phylum Chloroflexi and the family Amb-16S-1323 from the Hyphomicrobiales order. There was an increase in the Chloroflexi bacteria in samples on limestone and the Amb-16S-1323 group was more abundant off limestone.

Discussion

We assessed how spatial distance (geography) and the presence of CaCO3-rich rock (geology) are associated with variation in Oreohelix gut microbiome composition. By comparing environmental soil samples to the gut microbiomes of Oreohelix land snails, we also evaluated the extent of host filtering in host microbial community composition. We found that geology consistently affects snail gut microbiome composition, but a relatively conserved core set of taxa remained unchanged across snail samples. Specifically, the amplicon sequence variants (ASVs) in the families Trueperaceae and Spirosomaceae were found to be core snail gut microbes. Our results, along with past research, suggest that host filtering favors a specific set of microbes and that there is a small but significant effect of CaCO3 rock presence in the substrate used by snails on gut microbiome compositions for Oreohelix.

Differences between host and environment

We found that the host snails’ microbiomes were significantly different from that of soil samples. There were differences in alpha diversity, dispersion, and composition between sample types. Out of the alpha diversity measures, only Shannon’s diversity had high detection power so unsurprisingly, only Shannon’s diversity between sample types was shown to be significantly different, with soil diversity being higher. The similarity of snail microbiome richness to soils, known to be one of the richest reservoirs of microbial diversity, mirrors the results from previous work that shows that Oreohelix has elevated richness compared to other land snail taxa (Chalifour & Li, 2021). The dominant taxa were different between these samples. While they shared the dominant Phyla Proteobacteria and Actinobacteria, soils contained more Thermoproteota (an Archaea), and snails contained more Bacteriodota and Deinococcota. The differences between the composition of snail samples and their environment were also reflected in the results from differential abundance. We found many differently abundant taxa between sample types. Recent studies have also shown differences between Oreohelix host and surrounding environment, with snail gut samples more closely resembling those from the plant phyllosphere (Chalifour & Li, 2021; O’Rorke et al., 2015). These studies also found that some of the same bacterial families, Sphingobacteriaceae, Sphingomonadaceae, and Trueperaceae, dominated the snail microbiomes from populations across the large native range (Chalifour, Elder & Li, 2023). Lastly, Francisellaceae is extremely abundant in a few snail samples, spanning multiple sites, but not found in over half of the samples. This family appears to become successful when present but is not ubiquitous. There are known endoparasites within this family (Duron & Gottlieb, 2020), but more work needs to be done to identify if there is a meaningful relationship between this microbe and the snail host. While we decided not to speculate on the function of specific microbes using the present dataset, future work will aim to identify the sources of these microbes from phyllosphere and detritus samples, and understand the function of core taxa (present in all samples and relatively abundant) in the snails, such as Trueperaceae.

Effects of CaCO3 rock on microbial communities

We found that the presence of CaCO3 rock had a small and significant effect on the compositional differences of snail gut microbiomes. In soils, however, we found no effect of CaCO3 rock on compositional differences. We also found that neither sample type showed a difference in alpha diversity measures based on CaCO3 rock presence, though our detection power was low for those tests. Furthermore, geographic distance correlated with microbial dissimilarity in soils, but not snail samples. This could be due to dispersal limitation of soil microbes or another factor that changes spatially but was not measured in this study. We found that taxonomy (for snails), geographic distance (for soils), and site (for both) were important factors driving differences between samples. The sampling design of our study limited the inferences we can make about the effects of host taxonomy and site variables. Host taxonomic differences could be caused by differences in diet, habitat use, or differences in vertical transmission while site differences could be due to an even larger variety of factors including vegetation, aridity, and population structure for available environmental microbes. Each of these factors could be valuable avenues for future work to understand how the environment is affecting microbial community assembly in the snail gut. Previous work in Bornean microsnails living on limestone outcrops (Hendriks et al., 2021) found a link between host diet diversity, host diversity, and microbiome diversity. Resource availability and distance to caves where the snails are from were found to have significant effects on the snail microbiome diversity. Further work in similar land snail systems is needed to identify and study key microbes involved in host-habitat relationships, which could enable meaningful comparisons in microbial taxonomy. The differences in microbiomes between snails on and off CaCO3 rock could reflect functional needs of snails in their habitat or important host filtering. We would like to note that recent work by Chalifour, Elder & Li (2022) found that the process of snail tissue preservation can have a significant compositional impact on the microbiome, with relative increases in Enterobacteriaceae in preserved samples compared to fresh ones. Interestingly, they also show that after the initial preservation process, tissue stored in ethanol preserved a stable microbiome composition for decades. In the present study we did not test for the effects of preservation on Oreohelix microbiomes, as we processed the samples before (Chalifour, Elder & Li, 2022) was published. We also did not observe high relative abundances for Enterobacteriaceae and therefore cannot make any claims regarding these effects.

The finding that the gut microbiome of snails were significantly impacted by the presence of CaCO3 rock but the soil microbial communities were not could point to a dosage effect driven by the snails’ rasping (feeding by scraping the substrate with their radula) of the CaCO3 rock surface. For example, perhaps relatively little CaCO3 is being eroded from weather into soils and we know that snails are actively rasping the rock surface during their waking hours and likely ingest more of the substrate than would be leached into the soils. Captive breeding, calcium amendments, and transcriptomics could be used to provide more insight into any functional changes or dosage effect that CaCO3 rock might have on these microbial communities.

Conclusions

In our study, we used Oreohelix land snail gut tissue and adjacent soils to assess the effects of an abiotic resource, calcium-carbonate (via the presence of CaCO3-rich rock), on the communities of bacteria and archaea found in these samples. We found that the presence of CaCO3 rocks at a site had a significant effect on Oreohelix land snail gut but not on the soil microbial beta-diversity. Together with previous research, we have also identified what are likely many of the microbes that are important to Oreohelix and found that the dominant bacteria found in these snails are common across multiple studies from multiple labs (Chalifour & Li, 2021; Chalifour, Elder & Li, 2022; Chalifour, Elder & Li, 2023), adding support to these findings. Based on these results, it appears that these snails may have some important functional relationships to members of their microbiome or have relatively uniform host filtering processes. This stability is not expected based on previous research on invertebrate hosts (Hammer, Sanders & Fierer, 2019), but this might reflect the limited attention given to microbiomes of invertebrate animals. Our results also suggest that calcium availability is not correlated with the composition of soil microbial communities. In addition, we found two microbial families, Sandaracinaceae and Hyphomonadaceae, that were relatively more abundant in snails found off limestone. In particular, Sandaracinaceae was also found to be differentially abundant off limestone in the data subset (two paired locations with the same species on/off limestone). Other studies of snails on limestone outcroppings do not specifically mention the two taxa we found to be differentially abundant. In our dataset, their abundances are low, so they would not appear in the ‘core microbes’ lists typically reported in other studies. They emerge as important here because of our specific comparison between on- and off-limestone microbiomes While understanding the exact mechanism of this interaction was beyond the scope of this study, this lays the ground for future work. Future studies should investigate the functionality of the differently abundant microbes (from the snail vs. soil comparison) and utilize captive breeding programs with calcium carbonate treatments to assess whether there is a direct relationship between treatment and compositional changes in the microbiome. Finally, while we did see significant differences between snails on and off CaCO3 rock, we recognize that it may only be a small part of the bigger picture for directing microbial community assembly in Oreohelix land snails. It would be prudent to look at other factors such as diet, disease burden, ground cover, parasite load, and other factors that are important to host health and ecology and may, in turn, be important for the microbial communities within.

Supplemental Information

Supplemental Information 1 Supplementary Figures

Many thanks to the following members of the labs of Dr. Christine Parent and Dr. Luke Harmon for all the help with editing and conceptualization: Kristen Martinet, David Sneddon, Jane Dostart, and John Phillips. Finally, I. Oiler is grateful to the members of his graduate committee, Dr. Michael Strickland and Dr. Benjamin Ridenhour, for discussion and for providing comments on this manuscript. We would also like to acknowledge that this work was primarily done on the historic lands of the Nimiipuu (Nez Pierce) tribe.

Additional Information and Declarations

Competing Interests

Author Contributions

Field Study Permissions

Data Availability

The authors declare there are no competing interests.

Ian M. Oiler conceived and designed the experiments, performed the experiments, analyzed the data, prepared figures and/or tables, authored or reviewed drafts of the article, and approved the final draft.

T. Mason Linscott conceived and designed the experiments, performed the experiments, authored or reviewed drafts of the article, and approved the final draft.

Christine E. Parent conceived and designed the experiments, authored or reviewed drafts of the article, and approved the final draft.

The following information was supplied relating to field study approvals (i.e., approving body and any reference numbers):

No permits were required for these collections.

The following information was supplied regarding data availability:

The raw sequences are available at NCBI’s Sequence Read Archive: PRJNA1154269.

REVIEWER LINK for Figshare: https://figshare.com/s/8a0ba44b5acf59ec273d

The Idaho Oreohelix Microbiome files are available at Figshare: Oiler, Ian (2024). Idaho Oreohelix Microbiome Files. figshare. Dataset. https://doi.org/10.6084/m9.figshare.26972539.v1.

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
