# Peer review of "Geology correlates with gut microbial community composition in the Mountainsnails (Oreohelicidae: Oreohelix)"

_PeerJ, doi:10.7717/peerj.20080_

## Round 0.1 · original submission · Major Revisions

Dear Authors,

After review, I must inform you that substantial revisions are required before the manuscript can be considered for acceptance. While I appreciate the effort you have put into the submission, several key issues remain that need to be addressed.

Thank you for submitting your manuscript to PeerJ, and I look forward to receiving your revised version. Should you have any questions or require clarification during the revision process, please do not hesitate to reach out.

Sincerely,
Armando Sunny

Reviewer 1 ·

Basic reporting

This manuscript investigated how calcium-rich rocks and geographic distribution impacted microbiome composition of the soil and snails associated with those habitats. While the research question is interesting and the existing data analyses are rigorous, I found the manuscript in need of significant rewriting.

Language and Writing: Firstly, the manuscript needs polishing throughout, as I found it difficult to read in some sections. Many sentences feel vague or wordy, and some are not clearly relevant to the theme of the research. For instance:

Line 49: "From a human health perspective, a better understanding of the microbial communities of the world is important as reservoirs of pathogens, medicines, energy sources, and sustainers of biodiversity." This sentence feels disconnected from the main theme of the study and could be rewritten for clarity. Additionally, the phrasing is somewhat unclear. Perhaps focusing more on the relevance of microbial communities in the context of your study would improve flow.

Line 55: "To reach this goal, researchers must increase the breadth of studied host species to show how filtering of microbes by the host can interact with factors in the environment to affect the composition of the microbial communities differently. There is no explanation of the concept of host-filtering, and the sentence is confusing. What do you mean by "filtering of microbes by the host can interact with factors in the environment" and what do you mean by "affect the composition of the microbial communities differently"? Consider explaining 'host-filtering' in more detail and providing a specific example of how host and environmental factors interact to influence microbial composition.

Line 58: "Many factors can affect the composition of the microbes that exist in a given environment." Here, "that exist" is unnecessary and could be removed to make the sentence more concise.

Line 63: "For example, studies have shown that some important factors, like host phylogenetic relatedness, do not predict non-mammal gut microbiome similarity as well as in mammals". The use of "as well as in mammals" might confuse readers about what is being compared.

These are just a few examples from the first two paragraphs, but the entire manuscript would benefit from a thorough revision to ensure clarity, conciseness, and relevance to the central research theme.

Content and Structure: Additionally, the introduction could be expanded to better explain how calcium carbonate is expected to impact the snail microbiome. Currently, it is only stated that calcium carbonate is used for soil PH manipulation to "improve" microbial community. It is unclear how this is related to snail gut microbiome. Also:

Line 103: What does "shell trait expression" mean?

Line 105: What "distinct functional benefits" are being implicated here?

Overall, the introduction needs to be significantly revised so the ecological importance of host-filtering is well defined and discussed, and a logical link between soil calcium carbonate and animal gut microbiome composition is established.

Experimental design

The study has a small sample size, but the authors provided effective size tests and stated their analyses were conservative. So, sample size doesn't appear to be a major issue. What is more concerning is the inclusion of multiple species in the analyses, while not clearly stating out what the species are. Also, in the text, the snails were sometimes referred to as populations, indicating they are the same species, but sometimes referred to as different species/subspecies. I am not sure how to interpret the results until the snail identify information is clearly presented. For example, how many species are there? What are they? How many populations per species were included in the study? How many species were present at each site? Do they have different feeding strategies and mobility? Are all of them present on both calcium carbonate-rich and non-calcium carbonate rocks? Including a detailed description or a summary table that outlines these aspects would significantly improve the clarity of the study and help readers better understand the context of the findings.

Validity of the findings

As stated above, it is difficult to fully interpret the results without knowing species composition in this study. Additionally, many of the taxon-specific discussions focused on the phyla and family level (For example, when comparing soil and snail gut microbiome compositions). This doesn't seem to be very informative about the potential functional roles played by the different microbial communities. Is it possible to go to the genus level and provide more discussions on those groups?

Reviewer 2 ·

Basic reporting

The manuscript is written in clear and professional English and sufficient field background is provided. I only feel that the statement in lines 61-63 is a bit outdated. There are many studies done on other model and non-model organisms as well (e.g. Drosophila, Daphnia, C. elegans) and also studies in snails (such as Hendriks et al, 2021 in Ecology).
The words geology and geography are not well suited for the study as it seems that geology only implies the presence of calcareous rock and geography the distance between sites. Please rephrase this. Also, do you mean ASVs with features or what is meant with "features"?

The structure is fine, but the figures and tables may be improved.
Fig. 1: Please make the difference between the light and dark blue stronger and add the sampling location in the legend. Can you also add the full names and their abbreviations in the legend (given that you mention Rapid River, Lucile and Pittsburg Landing in the text).
Table 1: Under formula you placed the variables that were in the formula, but not the entire formula, so change the column name.

Besides, I may have missed it, but it seems that no raw data is shared.

Experimental design

The research question is well defined, but the methods could be improved and some necessary information is missing.

The geological characteristics and which information you obtained from previous studies to characterize sites in line 117 is unclear. Also, it is unclear what is meant with “opportunistically”, could you please add what was the minimum number of individuals found at a specific location.

In line 125 is mentioned that each snail species/subspecies are morphologically and geographically distinct and identified following previous research using the same populations. Did you have a different species at each site or were the paired ones in Rapid River, Lucile and Pittsburg Landing the same species/subspecies? Species is also used as a variable in some analysis, but it is unclear which species and where different species were found. Do you have any idea about the phylogenetic relationship between species and how this relates to their gut microbiota?

It seems that snails were without any liquid for two days before something was done with them, is that correct? At which temperature were they kept and did you check whether all snails were still alive before the dissection? Why did you add ethanol to the dissected guts that were stored in the freezer, this probably influenced the microbiota. Did you compare a fresh DNA extraction of the intestine with the stored ones in 95% ethanol? What is the reason you stored the intestine in ethanol until extraction?

No information is provided on the filtering of the data or on the quality of the reads that was obtained (which thresholds were chosen?). In line 139 is mentioned that pooling was done based on qubit scores, but it is unclear what is pooled and what these scores were exactly. Did you not sequence blank DNA extractions (and PCR solutions) to remove contaminants from your data? Without it is very hard to make conclusions about found ASVs as they might as well be contaminants. There is no information provided on rarefaction curves to ensure that samples reached plateau phase and neither on the normalization of the data. Did you check whether higher numbers of reads correlate with higher numbers of ASVs?

Validity of the findings

The significant values you are reporting from the PERMANOVAs can be due to the significant differences in dispersion, but there is only a significant difference in dispersion between soil and snail samples mentioned. Did you also use PERMDISP for the difference within snail samples and within soil samples? Can you please report those results to make sure that the differences found in PERMANOVAs are not purely driven by differences in dispersions.
Did you also test the partial Mantel test including geology and geography in the soil? Can you report those results.

When looking at differences between soil and snail samples for alpha diversity, did you take the different sites into account? Can you add a figure of alpha diversity in the supplementary information to show this significant difference for Shannon’s index that is not found in species richness and Faith’s PD.

It would be good to see the results of a blank DNA extraction as well, because it might be that this omnipresent ASV is also present in the control samples (or were the abundances of this ASV high in the snail samples?). If you have the ASVs of the blank, you could run something like decontam in R.

The conclusion in line 238 is strong and not really following from the results in the study. Are you for instance sure whether the influence of the rock is direct or indirect through the diet? Could it be that the diet is different and hence the bacteria?

Additional comments

Some extra comments:
L116: Specify how many samples.
L140: Please add a point after “intensity”.
L160: Please specify the other variables.
L170: Which habitat variables are considered? Did you also add snail species as a variable or was every site another species?
L260: What do you mean with the endoparasites, which ones?

Reviewer 3 ·

Basic reporting

The basic reporting is reasonable and adequate generally. I had the following comments:

Line 1 Change (genus: Oreohelix) to (Oreohelicidae: Oreohelix) to conform with common presentation.

Line 49 What literature supports gut microbiomes as sources of energy production vis a vis human health?

Line 61 “However, much of this research has been done in humans and non-human mammals which leaves a considerable taxonomic blind spot for understanding the mechanisms of gut microbial community dynamics across the entire tree of life.” While human studies are abundant, the literature is full of non-mammalian gut microbiome information. Consider rewording this statement or making it accurate.

Line 116 Include some basic locality information so the readers don’t have to use the map figure if they don’t want to. Also, the site names in the text are not those found on Figure 1.

Experimental design

The general experimental design is appropriate for this type of paper/project. Specific comments below.

Line 125 Are there genetic differences between the different snail species/subspecies? If not, that needs to be said. If so, then that relatedness needs to be addressed in the analysis.

Line 155 What is the rationale for using the outdated and less sampled Greengenes2 database when pre-compiled SILVA taxonomic classifiers are available?

Line 168 The results will make more sense to the reader if they are presented in the same order they are described in the methods.

Line 198 How do you distinguish between non-significant differences due to power issues versus ‘true’ non-significance?

Validity of the findings

I have no issue with the validity of the results themselves. It is a standard descriptive study found in the literature. I do, however, have issue with the wording of the findings in the discussion. The authors state in the their discussion and conclusion that they used Oreohelix gut tissue and adjacent soils to assess the effects of an abiotic resource, calcium-carbonate (via CaCO3-rich rock presence), on the communities of bacteria and archaea found in these samples. Unfortunately this is not accurate.

No portion of the project assessed any effect of calcium carbonate on the gut microbiomes. Rather, they assessed a correlative relationship between compositional differences and an environmental factor. A causal relationship was not established. It is also unclear how different the on and off sites were in terms of calcium carbonate concentration and availability. The difference between on and off carbonate rock could be explained by other abiotic factors (soil drainage, soil composition, temperature). What the authors present is that there are differences in two arbitrarily defined groups of sites in a single area that may (due to power issues) correlate with calcium carbonate. Any discussion beyond that is speculative at best and weakens their argument.

Again, as a simple descriptive study the manuscript is fine. However, the discussion and conclusions are incompatible with the data gathered and presented.

Additional comments

The authors designed a decent study but completely over-reached on their conclusions and cause-effect relationships. The project as presented is simply descriptive, which may or may not be appropriate for PeerJ. I read these types of descriptive projects in more regional or taxon-limited journals, and is where I would have submitted it.

---

## Round 0.2 · Major Revisions

Dear authors,

I hope this message finds you well. I have received Reviewer 1's comments on your manuscript titled "Geology shifts gut microbial community composition in wild Mountainsnails (genus: Oreohelix)". While your work is interesting, Reviewer 1 has raised significant concerns regarding the study design, particularly about the confounding effects of host phylogeny on microbiome variation.

To address these concerns, it is essential that you revise the study design, particularly in terms of how phylogenetic effects were controlled or accounted for in your analysis.

Due to these issues, major revisions are suggested before the manuscript can be reconsidered for publication.

Best regards,
Armando Sunny

Reviewer 1 ·

Basic reporting

This manuscript investigates how calcium-rich rock substrates and geographic distribution influence microbiome composition in both soil and associated Oreohelix land snails. The authors have made several meaningful revisions, including clarifying their methods, providing updated text, and specifying the identity of the snail species and subspecies included in the study. I appreciate the effort put into these revisions.

However, the inclusion of the phylogenetic tree in the revised version raises additional concerns about the study design, particularly regarding the confounding effects of host phylogeny on microbiome variation (detailed below). Furthermore, many portions of the manuscript would benefit from more careful editing to improve clarity, organization, and consistency in language.

Title: The current title may be somewhat misleading, as it implies a causal effect—that calcium-rich rock shifts the snail gut microbiome. However, the study design is observational and examines calcium-rich rock as one of several factors associated with variation in microbiome composition. I recommend revising the title to more accurately reflect the correlational nature of the findings and avoid overstating the influence of geology on gut microbiome.

Introduction: The revised introduction better aligns with the central theme of the study—how geological context may influence snail-associated microbiomes. The added content helps situate the research within broader ecological and microbiome literature. However, some portions of the text would benefit from further revision to improve clarity and narrative flow.

Line 60: For instance, the clause “between host evolutionary distinctness as measured by phylogenetic relatedness” is somewhat cumbersome; consider simplifying this to “host evolutionary divergence, measured by phylogenetic relatedness” for greater readability. Additionally, the sentence “the taxonomic blind spot for understanding the mechanisms of gut microbial community dynamics across the entire tree of life is ever shrinking” is somewhat informal for a scientific manuscript. You might consider rephrasing it to something more direct, such as “the gap in our understanding of gut microbiome dynamics across diverse taxa is narrowing.”

The transition from mammalian systems to non-mammalian systems could also be strengthened. As written, it is not entirely clear how the example given (i.e., host phylogenetic relatedness not predicting similarity in non-mammals) connects to the broader argument. You might briefly state why this finding is important.

Line 80: The statement “we can contrast this with taxa that are simply microbial grazers…” could be clarified. It's not immediately clear whether "this" refers to dietary reliance on gut microbes, or to the presence of symbionts. Additionally, the sentence “The varied processes that lead to how host factors (i.e. host filtering) affect microbial community assembly…” is slightly awkwardly phrased. Consider simplifying to “Recent work in microbial ecology has explored how host traits shape microbial community assembly through processes such as host filtering”.

Line 84: The final sentence introduces the study’s analytical approach, but its phrasing is vague: “Using a known association between a host animal and its environment our study analyzes whether this association is correlated with differences in host gut microbial composition.” I suggest rewording for precision.

Line 97: The connection between CaCO₃ substrate, pH, and gut microbiota could be better developed. Are the authors suggesting that snails living on CaCO₃-rich rocks acquire different microbes due to local environmental conditions, or that selection within the gut favors different microbial communities due to internal physiological changes (e.g., higher gut pH)? A clearer statement of the predicted mechanism would strengthen the logic of the study.

Experimental design

My major concern remains that the snails from different localities are different species/subspecies. The phylogeny showed that most sampled lineages fall within the Oreohelix strigosa species complex, but O. jugalis is more distantly related, raising concerns about the comparability of microbiomes across these taxa. As reported in the results, “snail species” is a strong and significant factor influencing gut microbiome composition. While the PERMANOVA models include species as a factor, this approach does not fully account for phylogenetic non-independence among host taxa.

Given that host species identity appears to be a major driver of microbiome variation, it becomes difficult to isolate the effect of CaCO₃ association from phylogenetic structure. Importantly, your dataset includes two lineages (Oreohelix from sites PITS01/01 and RR03/09) with populations both on and off CaCO₃-rich rocks. These represent ideal natural replicates for within-lineage comparisons where species identity is held constant. I strongly recommend conducting PERMANOVA (or related beta diversity analyses) separately within each of these lineages to assess the effect of substrate type in the absence of phylogenetic confounding.

Alternatively, incorporating a phylogenetic comparative approach would greatly strengthen the ecological inference regarding the role of CaCO₃.

Finally, I suggest moving the phylogeny from the Supplement to the main manuscript. It is essential context for interpreting microbiome patterns and assessing the extent of host divergence across sites and substrates.

Other issues in method:

Line 129: The phrase “a minimum of 3 and up to ten live snails opportunistically…” is vague. Please clarify the exact number collected per site if available, or state the mean and range.

Line 132. The sentence beginning “Each snail species/subspecies sampled…” is somewhat awkward. I recommend rephrasing for clarity, e.g.: “The snail species and subspecies sampled are morphologically and geographically distinct, and their identities were confirmed in prior work using these same populations [17].”

Additionally, consider briefly noting the number of taxa included and whether any are sympatric, as this could affect environmental exposure and microbiome overlap.

Line 167-169: The explanation of the beta diversity analyses is currently too brief. Please provide more detail on which factors were included—specifically, clarify whether beta diversity patterns were analyzed across localities, geology, host species, or other ecological variables. This clarification is essential for readers to interpret the scope and ecological relevance of the analyses.

Line 174-176: This sentence is difficult to follow as currently written. Please revise to clarify which variables or sample groups were being correlated. It is unclear whether the correlations refer to microbial composition, environmental variables, or both.

Line 177-179: It is unclear which comparisons were included in the differential abundance analyses. Please specify whether these were performed between snails from CaCO₃ vs. non-CaCO₃ substrates, between snail and soil samples, among species, or across sites.

Validity of the findings

Figure 2. Please revise the figure legend to clearly indicate what the colors and shapes represent

---

## Round 0.3 · Minor Revisions

Dear Authors,

Thank you for submitting the revised version of your manuscript titled “Geology correlates with gut microbial community composition in the Mountainsnails (Oreohelicidae: Oreohelix)” to PeerJ. We appreciate your thorough responses to the reviewers' comments and the improvements made throughout the manuscript.

After evaluating the revised version, we believe your study is nearly ready for acceptance. However, a few minor revisions are still required before we can proceed with final acceptance. Once these minor corrections are completed, we anticipate moving forward to formally accept your manuscript for publication. Please submit your revised manuscript and a brief response letter outlining the changes made. We look forward to receiving your final version soon.

Sincerely,
Armando Sunny

Reviewer 1 ·

Basic reporting

The authors did a good job addressing my previous comments. I have no more concerns.

Experimental design

All good.

Validity of the findings

Discussion and conclusion are robust.

Reviewer 2 ·

Basic reporting

Thank you for all the work and also for the revisions done. The manuscript is clear and professional English is used throughout. Although I am missing some literature on microbiomes in snails in the introduction as there is even research of other genera on limestone outcrops (e.g. Hendriks et al, 2021 in Ecology) and it would be nice to see whether there is overlap between the bacterial families found in this study and a former study. Because you are placing emphasis on the lack of research in the field of microbiomes in snails, it is more tempting to look for studies. The terms geology and geography are too broad for the current study and trying to overgeneralize. It is great you explained them, but I would remove those terms and use something like “carbonate rock proximity” and “spatial distance” instead as it is more correct.

Some minor points on reporting:
L45-47: The first sentence reads relatively interrupted due to all the brackets and it would be nicer if you could rephrase it more fluently such as: “Gut microbes can influence their hosts in many ways, including beneficial effects like enhancing host fitness, neutral interactions with no observable impact, and harmful outcomes such as reduced fitness.”
L141: Remove a point here.
L237: Remove “though this may be caused by the low power due to the size of this sample.” as that belongs to the discussion.

Experimental design

L150: It is a bit odd to me that you first dissect them and then place the dissected guts in 95% ethanol. Is this digestive tract not open or with potential cracks? You referred in your response (I was reviewer 2 in the first review round so my apologies to bring this up again) that this is common lab practice and that the same procedure has been followed in Chalifour et al. (2022). However, you can see in Fig. 3B, Fig. 4C, and Fig. 5 in Chalifour et al. (2022) that the fresh samples are often significantly different from the treated samples. I agree nothing can be changed anymore in the current study, but acknowledge this (in the discussion perhaps) and perhaps refer to this study in your methods.
L159: It is unclear to me what you mean with the pooling of the samples. Which samples were pooled? And do you mean before barcoding the samples or after? Or do you rather mean that you used qubit scores and gel band intensity as a way of normalizing to make sure that equal amounts of DNA per sample are present in the libraries? If that is what you mean, please make it clear.
L166: You have to mention the specific trimming threshold here for reproducibility.
L178: I would expect first to assign taxonomy and then to obtain alpha metrics. Regarding taxonomy, did you remove ASVs that were not classified? Did you look into read length and potentially removed reads that were too short or too long compared to the region of interest?

Validity of the findings

no comment

Additional comments

L266-268: Did you also find the following in the smaller dataset with only RR and PITS? “Within snail samples, we found that there were two families, Sandaracinaceae and Hyphomonadaceae, that were differently abundant between samples on CaCO3 rock and off.” Did you check if these were also found in other snail species on limestone outcrops in literature? It would be great to see whether this is a more general finding.

Reviewer 3 ·

Basic reporting

The basic reporting is reasonable and adequate generally. The authors have adequately addressed the reviewers comments and made the appropriate edits.

Experimental design

The general experimental design is appropriate for this type of paper/project. The authors have adequately addressed the reviewers comments and made the appropriate edits.

Validity of the findings

I have no issue with the validity of the results themselves. It is a standard descriptive study found in the literature. The authors did an excellent job addressing the reviewers comments and making the appropriate edits. I appreciate the clarifications and explanations they provided.

Additional comments

I would make two additional changes at this point.

1. Check the tense throughout the manuscript. Anything relating to the study directly should be in the past tense (we DID this, we SAW that, etc.). There are points in the methods and results where present-tense is used incorrectly.

2. There are portions of the results that are written as discussion. Keep the results factual and directly in line with the methods. Putting them in context etc. belongs in the discussion.

---

## Round 0.4 · accepted · Accept

Dear Authors,

I am pleased to inform you that your manuscript entitled “Geology correlates with gut microbial community composition in the Mountainsnails (Oreohelicidae: Oreohelix)” has been accepted for publication in PeerJ.

On behalf of PeerJ, I would like to congratulate you and your co-authors for this achievement. Your article will now move forward to the production stage, where it will be copyedited, typeset, and prepared for online publication. You will be contacted by the production team for any final queries during this process.

Thank you for choosing PeerJ as the venue for your work. We look forward to seeing your article published and to your continued contributions to the scientific community.

Sincerely,

Armando Sunny